# Tunable Optical Frequency Comb Generated Using Periodic Windows in a Laser and Its Application for Distance Measurement

**DOI:** 10.3390/s23218872

**Published:** 2023-10-31

**Authors:** Zhuqiu Chen, Can Fang, Yuxi Ruan, Yanguang Yu, Qinghua Guo, Jun Tong, Jiangtao Xi

**Affiliations:** School of Electrical, Computer and Telecommunications Engineering, Faculty of Engineering and Information Sciences, University of Wollongong, Northfields Ave, Wollongong, NSW 2522, Australia; zc606@uowmail.edu.au (Z.C.); cf861@uowmail.edu.au (C.F.); yruan@uow.edu.au (Y.R.); jtong@uow.edu.au (J.T.); jiangtao@uow.edu.au (J.X.)

**Keywords:** optical feedback, laser dynamics, laser diode, optical frequency comb, distance measurement

## Abstract

A novel method for the generation of an optical frequency comb (OFC) is presented. The proposed approach uses a laser diode with optical feedback and operating at a specific nonlinear dynamic state named periodic window. In this case, the laser spectrum exhibits a feature with a series of discrete, equally spaced frequency components, and the repetition rate can be flexibly adjusted by varying the system parameters (e.g., external cavity length), which can provide many potential applications. As an application example, a dual-OFC system for distance measurement is presented. The results demonstrate the system’s ability to achieve target distance detection, underscoring its potential for real-world applications in this field.

## 1. Introduction

In recent decades, the optical frequency comb (OFC) has seen significant advancements, providing a wide range of stabilized optical wavelengths that can be simultaneously utilized. By emitting a broad spectrum of uniformly spaced ultrashort pulses with stable and narrow lines, these OFCs effectively function as continuous-wave (CW) lasers evenly distributed across the frequency domain [1,2]. Capitalizing on these inherent advantages, OFCs have found applications in various fields, such as atomic clock comparisons [3], long distance quantum key distribution [4,5], calibration of astronomical spectrographs [6], ultralow-noise microwave generation [7], distance measurement and laser ranging [1,8], as well as many applications in medical areas [9,10]. In the context of distance measurement, several OFC-based techniques have been proposed. These include time-of-flight (TOF) distance measurements, which are achieved by the optical cross-correlation method [11]; the synthetic wavelength interferometry method, which generates harmonics of an OFC’s repetition rate through different optical modes and enables a connection between the peak-finding method and single-wavelength heterodyne interferometry [12]; the multiple-wavelength interferometry method, which is based on multichannel phase detection to determine the target position [13]; and others.

However, in most of these ranging applications, single-frequency combs are employed, but they do not fully exploit all the teeth of the comb. This limitation arises because it is often challenging to distinguish adjacent comb lines with megahertz to gigahertz mode spacing [14]. Additionally, the process of adjusting the repetition rate to align the reference pulses and target pulses, which is essential for single-comb ranging, introduces a significant complexity to the systems [15,16]. To address these limitations, dual-comb interferometry is utilized, employing two optical frequency combs with slightly different repetition rates. The beams from these two OFCs interfere on a photodetector, generating a microwave frequency comb (MFC) in the beat spectra. This enables the acquisition of intensity and phase information encoded in the individual comb modes through standard high-speed electronics, facilitating a fast Fourier transform analysis [17]. This principle considerably simplifies the receiver setup and has been demonstrated to be effective for absolute distance measurements, aiming to achieve improved ranging resolution, high update rate, and a low signal-to-noise ratio [18,19].

Regarding the generation source of the dual-comb signal, various methods have been developed. In one approach [20], mode-locked lasers (MLLs) with slightly different repetition rates are utilized for distance measurement. This results in a comblike spectrum through the interaction between the gain medium (erbium-doped fiber) and the laser resonator cavity. However, these systems are costly and complex, requiring additional devices like Kalman filters to mitigate timing jitter. Another method [21] involves the generation of two optical comb signals through electro-optic modulation for achieving ranging. This approach increases system complexity as each electro-optic modulation comb generation scheme consists of an intensity modulator followed by a phase modulator. Additionally, in [22], two femtosecond fiber lasers are used for achieving absolute distance measurement, involving the synchronization of multiple laser modes to produce a train of optical pulses with a fixed repetition rate. Nonetheless, it should be noted that femtosecond lasers can be relatively expensive, potentially leading to increased system costs.

Furthermore, an essential approach to optical frequency comb (OFC) generation involves leveraging laser dynamics triggered by external perturbations. The dynamics of a laser diode (LD) with external perturbation can be highly intricate, with interactions between relaxation oscillation and external feedback frequencies leading to nonlinear instabilities. The method of optical frequency comb generation grounded in laser nonlinear dynamics offers versatile advantages by enabling a range of nonlinear dynamics, and by manipulating controllable parameters such as optical feedback strength, the LD can exhibit various dynamic states, including steady state, P1 oscillation, quasi-periodic oscillation, and chaos, providing flexibility for diverse applications [23,24]. Additionally, it allows a precise control over the comb properties, including repetition rate and comb teeth relative amplitudes, enabling tailored comb solutions for various needs [25,26,27]. Optical injection is a technique that introduces an external perturbation to the LD by injecting light from another LD. In previous studies [25,26], an OFC was generated by optically injecting an LD, leading to nonlinear dynamics in the period-one (P1) oscillation state. The repetition rate of the OFC could be tuned using electro-optic modulation. Additionally, in another study [27], an OFC was generated by injecting a suitable OFC into an LD, resulting in a rich variety of nonlinear dynamics that were characterized by a comblike behavior. The number of comb lines and relative amplitudes of these lines could be adjusted by controlling injection parameters, such as injection strength and detuning frequency. Optical injection enables an adjustable repetition rate for flexibility, while providing stable, low-noise microwave signals, ensuring precision and stability in OFC generation. However, it is essential to recognize that the optical injection method introduces complexities, necessitating the use of at least two LDs, which, in turn, results in higher costs and implementation difficulties.

The compact system configuration of an LD with external optical feedback (EOF) makes it a highly suitable candidate for investigating the properties of LD nonlinear dynamics. This is achieved by introducing external perturbation through the reflection and partial reinjection of emitted light back into the laser cavity. Its practicality and efficiency in providing such perturbations are widely acknowledged [28,29,30].

In this study, we investigate the use of external optical feedback (EOF) to induce laser dynamics for the generation of optical frequency combs (OFCs) with tunable repetition rates. Unlike conventional methods, our approach involves operating a laser diode (LD) within periodic windows, resulting in an OFC source offering a wide tunable repetition rate range of up to 18.07 MHz by adjusting the external cavity length. This approach simplifies the setup significantly, as it relies on a single laser diode with optical feedback, eliminating the need for complex configurations involving mode-locked lasers, electro-optic modulators, and optical amplifiers. Moreover, it offers cost-effectiveness compared to traditional dual-comb systems that rely on expensive mode-locked lasers. By utilizing specific periodic window regions, our method produces stable and low-noise optical frequency combs, addressing common stability challenges encountered in laser systems. Additionally, our technique provides precise control over comb parameters, such as repetition rate and the number of comb teeth, through adjustments in the external cavity length and other operating conditions—a level of controllability challenging to achieve in most conventional comb generation methods. These attributes highlight the potential of our approach for practical applications in various fields, including distance measurement.

## 2. Periodic Window and OFC Generation

### 2.1. Dynamic States of an LD with EOF

Figure 1 illustrates the schematic diagram of the LD with EOF used in our study. The system comprises an LD and its external cavity. A reflection mirror positioned on a piezoelectric transducer (PZT) is used to form the cavity. The injection current ratio to the LD is precisely controlled by the LD controller. A variable attenuator (VA) is inserted into the cavity to adjust the EOF strength. The PZT controller governs the amplitude and frequency of the PZT for a precise adjustment of the external cavity length.

To study the laser dynamic states, one portion of the light is directed to an optical spectrum analyzer (OSA) for spectrum analysis. The other portion is captured by a photodiode (PD) and displayed on an oscilloscope (OSC) for a time domain analysis. An optical isolator (OI) is used to prevent backscattering of the light from beam splitter 2 ($BS_{2}$).

The nonlinear dynamics of a laser diode (LD) with EOF are described by the widely recognized Lang and Kobayashi (L-K) equations, as shown by Equations (Equation 1)–(Equation 3) [31]. E(t) represents the amplitude of the electric field, ϕ(t) corresponds to the electric field phase, and N(t) denotes the carrier density. In this paper, $E^{2}\left(t\right)$ is treated as the laser intensity.
(1)$$\begin{array}{cc} \frac{dE(t)}{dt}= & \frac{1}{2}\left\{G\left[N(t),E(t)\right]-\frac{1}{\tau_{p}}\right\}E\left(t\right)+\frac{\kappa }{\tau_{in}}\cdot E\left(t-\tau \right)\cdot \cos\left[\omega_{0}\tau +\phi \left(t\right)-\phi \left(t-\tau \right)\right] \end{array}$$
(2)$$\begin{array}{cc} \frac{d\phi (t)}{dt}= & \frac{1}{2}\alpha \left\{G\left[N(t),E(t)\right]-\frac{1}{\tau_{p}}\right\}-\frac{\kappa }{\tau_{in}}\cdot \frac{E(t-\tau )}{E(t)}\cdot \sin\left[\omega_{0}\tau +\phi \left(t\right)-\phi \left(t-\tau \right)\right] \end{array}$$
(3)$$\begin{array}{c} \frac{dN(t)}{dt}=\frac{J}{eV}-\frac{N(t)}{\tau_{s}}-G\left[N(t),E(t)\right]E^{2}\left(t\right) \end{array}$$
where $G\left[N(t),E(t)\right]=G_{N}\left[N\left(t\right)-N_{0}\right]\left[1-\varepsilon \Gamma E^{2}\left(t\right)\right]$ is the modal gain per unit time. The physical meanings and values of parameters used in Equations (Equation 1)–(Equation 3) are shown in Table A1 in Appendix A [32].

The parameters relevant to the LD used for our study on OFC generation are presented in Table A1 (called internal parameters). The other parameters including injection current, optical feedback strength, and external cavity length are also shown in Table A1 (called controllable parameters). The external parameters are controllable and serve to govern the dynamic states of the LD. An LD with EOF may operate at different states including steady state (S), P1 oscillation state, quasi-periodic (QP) oscillation state, periodic windows state, and chaos. Regarding periodic window, it is the region embedded within chaos and demonstrating a quasiperiodic behavior [23,33,34].

Note that period-N (denoted by PN) in the following means there are N peaks within a period of the laser intensity in time domain. Each state can be identified based on the waveform of $E^{2}\left(t\right)$. The transition route from a steady state to chaos, influenced by one or more controllable variables, is known as the bifurcation diagram [23,24]. To construct the bifurcation diagram, we sampled the local maxima $E^{2}{\left(t\right)}_{Max}$ (or minima $E^{2}{\left(t\right)}_{Min}$) of the waveform $E^{2}\left(t\right)$ for a given set of LD controllable parameters. This allowed us to gain valuable insights into the complex dynamics and behavior exhibited by the LD system under different cavity conditions.

Based on our study, a specific set of controllable parameters were chosen as: α=2.0, L=5.2cm and $J=1.2J_{\mathrm{th}}$. We firstly explored the impact of the EOF strength (κ) on the dynamic states by varying κ over the range (0, 0.06). The corresponding bifurcation diagram was obtained and is depicted on Figure 2.

From Figure 2, we observe six dynamic states. The steady state corresponds to 0<κ<0.0147. As κ increases beyond this point, an undamped relaxation oscillation occurs after the Hopf bifurcation, leading the system to enter the period-1 (P1) oscillation state within the range of 0.0147<κ<0.0281. Subsequently, a quasi-periodic (QP) state emerges corresponding to 0.0281<κ<0.0318. The first chaotic region appears in the range of 0.0318<κ<0.0399 and the second chaotic region with 0.0456<κ<0.0600. Within the two chaotic regions, we observe a periodic window starting at κ=0.0399 and ending at κ=0.0456. This periodic window further exhibits a P3 state followed by a P6 state.

### 2.2. OFC Generation and Its Tunability

Let us examine the characteristics of the laser intensity $E^{2}\left(t\right)$ within this periodic window. We selected two operating points with κ=0.041 and κ=0.042, respectively, and plotted their corresponding time series (i.e., laser intensity $E^{2}\left(t\right)$) in Figure 3. Figure 3a shows that the laser intensity exhibits a periodic structure with three peaks within each period of 1.25 ns. Figure 3b shows a periodic structure with six peaks within each period of 2.48 ns. After extensive simulation tests, as a result, this periodic window combined a P3 state along with a P6 state with κ=0.0416 as their boundary line.

We then looked at the spectrum feature when the LD operating at the periodical window. Figure 4 illustrates the laser spectrum at κ=0.041 within the P3 region. Clearly it shows a series of discrete, equally spaced frequency components, demonstrating the generation of an OFC in this operating condition. The OFC generated exhibits a repetition rate of $f_{rep}=0.8$ GHz. The total bandwidth spans from 193.365 THz to 193.464 THz. Within the −10 dB region, there are 66 comb teeth. Hence, the introduced LD with external optical feedback (EOF) system can be regarded as an optical frequency comb (OFC) generation source. In the subsequent sections, we refer to it as OFC-EOF.

Next, we show that the repetition rate of the generated OFC can be adjusted by varying the cavity length *L*. By gradually increasing *L* with increments ΔL=15.50 nm, 23.25 nm, 31.00 nm and 38.75 nm, respectively, the repetition rate changes from 810.5 MHz to 818.4 MHz, as depicted in Figure 5.

To further determine the tunable range of the repetition rate denoted as $\Delta f_{rep}$, extensive simulations were conducted to generate an OFC under different external cavity lengths *L* within the range of 0≤ΔL≤37.2 nm. In these simulations, α and *J* were fixed at α=2.0 and $J=1.2J_{\mathrm{th}}$, and κ was fixed at κ=0.041.

As shown in Figure 6, the repetition rate of the OFC increased from 800.67 MHz to 818.74 MHz as *L* increased from 5.2 cm to 5.2 cm + 37.2 nm.

## 3. A Dual-OFC System for Distance Measurement

The simulation analysis revealed that the proposed LD system, operating within a specific periodic window region, could generate an optical frequency comb (OFC) source with a tunable repetition rate spanning from 800.67 MHz to 818.74 MHz. This tunability allowed the system to generate OFC signals with repetition rate differences of up to 18.07 MHz. Such a feature of OFC generation, coupled with its tunable range, holds the potential for various applications in optical and microwave measurements. As an application example, in this section, we present a dual-OFC system for achieve target distance detection. This application showcases the versatility and advantages of the proposed system in practical and precise distance measurement scenarios.

Figure 7 illustrates the schematic of the proposed dual-OFC system for distance measurement. The boxed part of the diagram represents the source of the dual-comb light, consisting of two independent OFC-EOF systems. These systems generate two distinct optical frequency combs, denoted as $\mathrm{OF}C_{1}$ and $\mathrm{OF}C_{2}$, with a slight difference in their repetition rates, referred to as $f_{rep\_diff}$. Each OFC-EOF system incorporates a laser diode (LD) with an external feedback cavity, and the cavity lengths ($L_{1}$ and $L_{2}$) are precisely adjusted using a piezoelectric transducer (PZT) with a difference of 1.55 nm. For a typical dual-OFC ranging system, generally, a single-mode DFB laser diode is used as the OFCs generation source, after amplification by EDFAs to provide an output power up to around 200–500 mW [16,22,35,36].

Optical isolators are used to prevent light backscattering, which could potentially interfere with the dynamic states of the LDs. The OFC generated by $\mathrm{OFC}-\mathrm{EO}F_{1}$ and $\mathrm{OFC}-\mathrm{EO}F_{2}$ are designated as the $\mathrm{OF}C_{1}$ beam and $\mathrm{OF}C_{2}$ beam, respectively. Upon exiting the system, the $\mathrm{OF}C_{1}$ beam is directed towards a beam splitter (BS) and subsequently divided into two beams: a target measurement beam denoted by $\mathrm{OF}C_{1\;\;\;\_\;\;\;\mathrm{mea}}$ (expressed by $E_{1\_mea}\left(t\right)=\sum_{n}E_{1n}e^{j\left[2\pi \left(nf_{1\_rep}+f_{1\_1}\right)t-\phi_{n}\right]}$) and a reference beam denoted by $\mathrm{OF}C_{1\;\;\;\_\;\;\;\mathrm{ref}}$ (expressed by $E_{1\_ref}\left(t\right)=\sum_{n}E_{1n}e^{j\left[2\pi \left(nf_{1\_rep}+f_{1\_1}\right)t\right]}$), where $E_{1\_mea}\left(t\right)$ and $E_{1\_ref}\left(t\right)$ are electric fields of the reflected measurement beam and reference comb, respectively. The $\mathrm{OF}C_{1\;\;\;\_\;\;\;\mathrm{mea}}$ beam and $\mathrm{OF}C_{1\;\;\;\_\;\;\;\mathrm{ref}}$ beam containing multiple optical frequencies components (up to *n*) have light magnitudes $E_{1n}$ and $E_{2n}$, respectively. These beams contain multiple optical frequency components with respective electric fields. The first comb tooth frequencies in $\mathrm{OF}C_{1\;\;\;\_\;\;\;\mathrm{mea}}$ and $\mathrm{OF}C_{1\;\;\;\_\;\;\;\mathrm{ref}}$ are $f_{1\_1}$, and both OFCs have the same repetition rate denoted as $f_{1\_rep}$. The phase difference between each pair of frequency comb teeth in $\mathrm{OF}C_{1\;\;\;\_\;\;\;\mathrm{mea}}$ and $\mathrm{OF}C_{1\;\;\;\_\;\;\;\mathrm{ref}}$ is represented by $\phi_{n}$.

The $\mathrm{OF}C_{1\;\_\;\mathrm{mea}}$ beam is directed to the target at a distance *D* and then reflected by the target surface. Simultaneously, the reference beam ($\mathrm{OF}C_{1\;\_\;\mathrm{ref}}$) is reflected by a reflector, carrying distance information in its phase. The measurement beam ($\mathrm{OF}C_{1\;\_\;\mathrm{ref}}$) and the reflected beam $\mathrm{OF}C_{1\;\_\;\mathrm{mea}}$ are recombined before passing through $\mathrm{CO}L_{2}$. The interference between the two beams is then amplified by an erbium-doped fiber application amplifier (EDFA) to compensate for any power losses during propagation. Subsequently, the interference beam is combined with the $\mathrm{OF}C_{2}$ beam (expressed by $E_{2}\left(t\right)=\sum_{n}E_{2n}e^{j\left[2\pi \left(nf_{2\_rep}+f_{2\_1}\right)t\right]}$) using a fiber coupler (FC) and detected by a photodiode (PD), yielding a photocurrent denoted as $I_{beat}$ (expressed as $I_{beat}={\left|E_{1\_mea}\left(t\right)+E_{1\_ref}\left(t\right)+E_{2}\left(t\right)\right|}^{2}$), where the first comb tooth frequency of $\mathrm{OF}C_{2}$ is denoted as $f_{2\_1}$, and its repetition rate is $f_{2\_rep}$. There exists a slight difference ($f_{rep\_diff}$) between $f_{1\_rep}$ and $f_{2\_rep}$, which can be controlled by adjusting the external cavity length $L_{1}$ and $L_{2}$. It is assumed that the oscillations of the internal resonant cavities of the LDs in $\mathrm{OFC}-\mathrm{EO}F_{1}$ and $\mathrm{OFC}-\mathrm{EO}F_{2}$ are synchronized, ensuring the equality of $f_{1\_1}$ and $f_{2\_1}$. After passing through a low-pass filter, we gain the output photocurrent denoted by $I_{beat\_low}$, expressed by $I_{beat\_low}=\sum_{n}E_{1n}E_{2n}\cos\left(\frac{\phi_{n}}{2}\right)\cos\left(2\pi nf_{rep\_diff}t-\frac{\phi_{n}}{2}\right)$, and it is recorded by the OSC for further signal processing to extract the distance.

By applying a fast Fourier transform (FFT) on $I_{beat\_low}$, the spectrum of the resulting beat spectrum is obtained. The repetition rate of the beat signal is equal to the repetition rate difference between $\mathrm{OF}C_{1\;\_\;\mathrm{mea}}$ and $\mathrm{OF}C_{1\;\_\;\mathrm{ref}}$. The distance information is then extracted from the phase of the respective frequency comb components. This configuration allows us to achieve target distance detection through the analysis of the beat spectrum.

The main process for retrieving the distance information can be found in [37]. Figure 8 is the processing diagram. As depicted in Figure 8, the process of distance measurement involves a sequence of six distinct steps. In the initial stage, the electric fields of $\mathrm{OF}C_{1\;\;\;\_\;\;\;\mathrm{mea}}$, $\mathrm{OF}C_{1\;\;\;\_\;\;\;\mathrm{ref}}$, and $\mathrm{OF}C_{2}$ are derived from the dual-OFC source. The mathematical representations of their respective electric fields are presented within the first block. Subsequently, in step 2, the $\mathrm{OF}C_{2}$ is subjected to a beating process with the $\mathrm{OF}C_{1\;\;\;\_\;\;\;\mathrm{mea}}$ and $\mathrm{OF}C_{1\;\;\;\_\;\;\;\mathrm{ref}}$ signals, which are reflected from the target and reflector, respectively. This interaction takes place on a PD. Moving on to step 3, the detected current signal denoted as $I_{beat}$ originating from the photodiode passes through a low-pass filter. The primary purpose of this filtering step is to eliminate high-frequency components. It is noteworthy that $I_{beat\_low}$ in this context symbolizes the superimposed intensity of a beat signal characterized by a repetition rate of $f_{rep\_diff}$. Subsequently, the collected spectrum of the low-pass-filtered signal is acquired using an OSC. Each comb tooth’s power is extracted individually, and the application of the fast Fourier transform (FFT) serves to eliminate direct current (DC) terms. During steps 4 and 5, the phase ($\phi_{n}$) pertaining to each comb tooth is determined by subjecting the low-pass-filtered signal to the Hilbert transform, where $\tau_{D}$ is the time delay between the $\mathrm{OF}C_{1\;\;\;\_\;\;\;\mathrm{mea}}$ beam and the $\mathrm{OF}C_{1\;\;\;\_\;\;\;\mathrm{ref}}$ beam, given by $\tau_{D}=2D/c$, where *D* is the distance difference between target and reflector, and *c* is the speed of light. The Hilbert transform is an operation that converts a real-valued signal into a complex-valued signal while preserving the original frequency components. It is commonly used in signal processing to extract the instantaneous amplitude and phase information from a signal [38].

Then, we employ an unwrapping process to determine the fit line slope of the relationship between phases and the order of a comb tooth (up to *n*). Lastly, in the sixth step, we compute the measured distance $D_{mea}$ by equations provided in the sixth block. The measured distance can be calculated by the first equation in the step 6 block when $0\leq \left|D_{pre}\right|\leq 1/2D_{namb}$, where $D_{namb}$ is the range of nonambiguity denoted by $D_{namb}=c/2f_{1\_rep}=187,200\;\mu$m. We observe that when the $D_{pre}$ is in the range of $0\leq \left|D_{pre}\right|\leq 1/2D_{namb}$, the $d\phi_{n}/dn$ (slope) can be used for calculating $D_{mea}$, indicating a positive correlation between $D_{pre}$ and $d\phi_{n}/dn$. However, as $D_{pre}$ increases, the $d\phi_{n}/dn$ becomes closer to π and then decreases when $D_{pre}$ is larger than $1/2D_{namb}$, falling into the range of $1/2D_{namb}\leq \left|D_{pre}\right|\leq D_{namb}$. In this case, $D_{pre}$ and $d\phi_{n}/dn$ change to be negatively correlated and the expression of $D_{mea}$ is modified to the second equation in the step 6 block. When $D_{mea}$ exceeds $D_{namb}$, the relationship between $D_{pre}$ and the slope of unwrapped phases repeats for each $D_{namb}$. To measure distances larger than $D_{namb}$, the repetition rate needs to be tuned accordingly. For a specific $D_{pre}$, different repetition rates can result in different $d\phi_{n}/dn$’s. In this case, $D_{mea}$ can be calculated by the third equation in the step 6 block. Following the aforementioned processing diagram, we present an example with a predefined distance of $D_{pre}$ = 2,400,000.0 μm. In this case, $D_{pre}$ exceeds $D_{namb}$, necessitating the use of the third equation in the step 6 block for distance measurement. $\mathrm{OFC}-\mathrm{EO}F_{1}$ and $\mathrm{OFC}-\mathrm{EO}F_{2}$ were both fixed with α=2.0, JRatio=1.2 and κ=0.041, resulting in the generation of two OFC signals with repetition rates of $f_{1\_rep}=$ 800.67 MHz and $f_{2\_rep}=f_{1\_rep}+$ 1.1267 MHz, respectively. In this case, an $f_{rep\_diff}$ of 1.1267 MHz was achieved.

Figure 9a shows the spectrum of the beat signal of the $\mathrm{OF}C_{1\;\;\;\_\;\;\;\mathrm{mea}}$ beam, $\mathrm{OF}C_{1\;\;\;\_\;\;\;\mathrm{ref}}$ beam, and $\mathrm{OF}C_{2}$ beam after low-pass filtering. It is observed as a 59-teeth beat spectrum with the first comb tooth frequency and repetition rate both at 1.1267 MHz, which is consistent with the expression of $I_{beat\_low}$ shown in the step 3 block. Subsequently, the power of each comb tooth was extracted. To extract the distance information carried by the phase of each comb tooth, we applied the Hilbert transform on the extracted power spectrum, as shown in Figure 9b. The phase of each comb tooth was wrapped within the range of (−π,π). To obtain the derivative of phase with respect to *n*, an unwrapping process was applied. The results of the 59 points are shown in Figure 9c, and the corresponding fitting line is plotted in black. It can be observed that the obtained $d\phi_{n}/dn$ was 1.1992.

In this study, our OFC-EOF system can generate OFCs with variable repetition rates ranging from 800.67 MHz to 818.74 MHz. In the context of distance measurement, the repetition rates $f_{1\_rep}$ and $f_{2\_rep}$ are tuned by precisely adjusting the external cavity lengths $L_{1}$ and $L_{2}$ in increments of ΔL=38.75 nm. In Figure 9d, the correlation between the varied repetition rate $f_{1\_rep}$ and the corresponding $d\phi_{n}/dn$ is illustrated. The distance can be obtained twice with the results of $D_{mea}$ = 2,400,032.5 μm and $D_{mea}$ = 2,400,070.3 μm, respectively.

## 4. Conclusions

In this study, we successfully demonstrated the generation of optical frequency combs (OFCs) with a tunable repetition rate using a laser diode (LD) with external optical feedback (EOF). By operating the LD within periodic windows, we were able to control the laser dynamics and achieve a tunable repetition rate for the OFCs. The bifurcation diagram provided insights into the different laser dynamic states, with a focus on the period-three window. We established the relationship between system parameters, such as the external cavity length and the tunable range of the repetition rate. Furthermore, we showcased the practicality of our proposed OFC generation method by implementing it in a dual-OFC system for distance measurement. The results showed a precise target distance detection, indicating the method’s potential for practical applications in fields such as remote sensing and precision metrology. Overall, this study contributes to the understanding of LD nonlinear dynamics and provides a novel approach for generating OFCs with tunable repetition rates. The tunability capabilities of the proposed method open up new possibilities for applications in various fields requiring precise frequency comb sources.

## Figures and Tables

**Figure 1 sensors-23-08872-f001:**
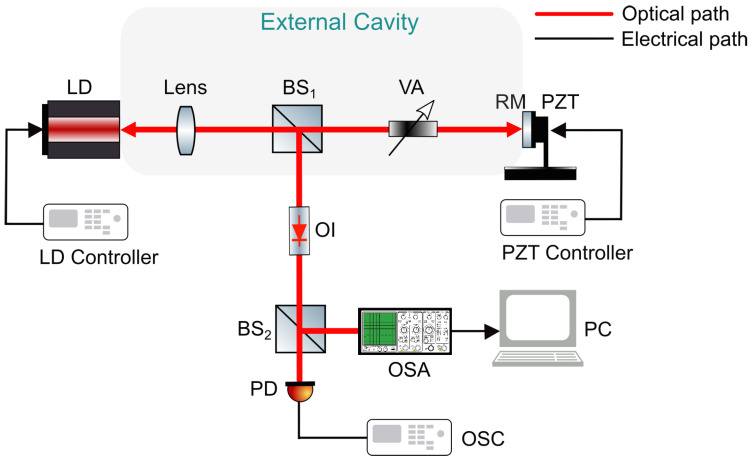
Schematic diagram for an LD with EOF. LD, laser diode; BS, beam splitter; VA, variable attenuator; RM, reflection mirror; PZT, piezoelectric transducer; OI, optical isolator; OSA, optical spectrum analyzer; PD, photodiode; OSC, oscilloscope; PC, computer.

**Figure 2 sensors-23-08872-f002:**
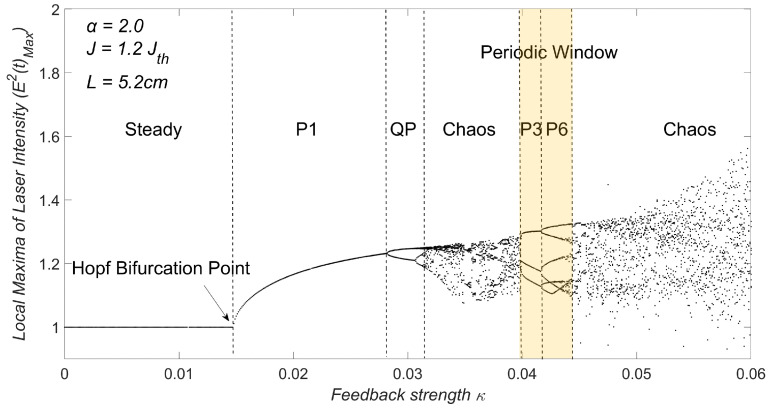
Bifurcation diagram with a periodic window embedded in chaos.

**Figure 3 sensors-23-08872-f003:**
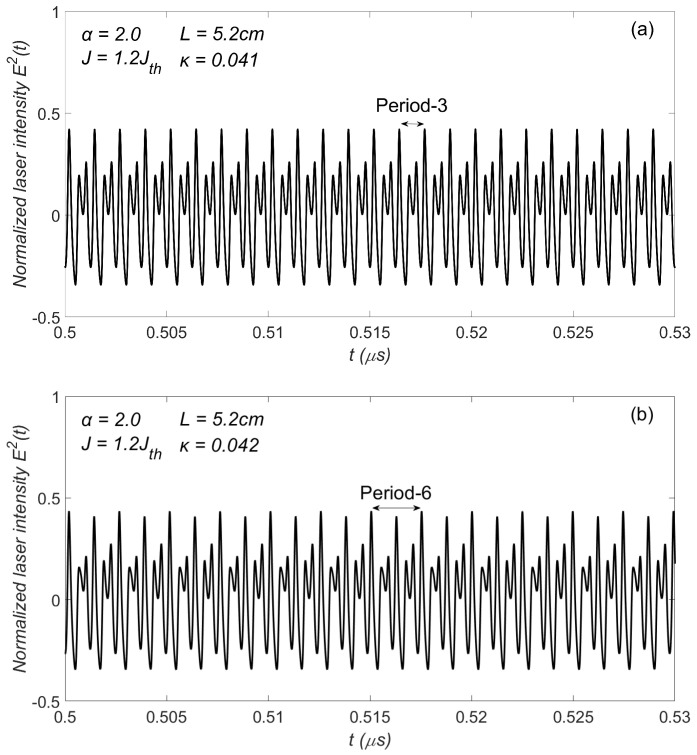
Time series (**a**) P3 region. (**b**) P6 region.

**Figure 4 sensors-23-08872-f004:**
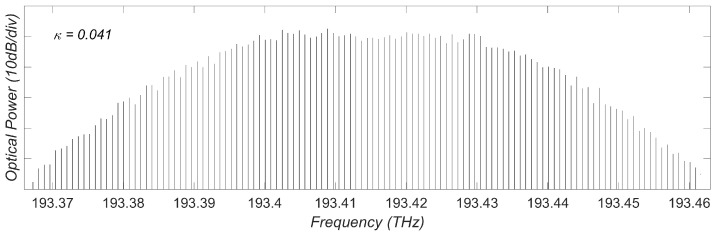
Optical spectrum at κ=0.041.

**Figure 5 sensors-23-08872-f005:**
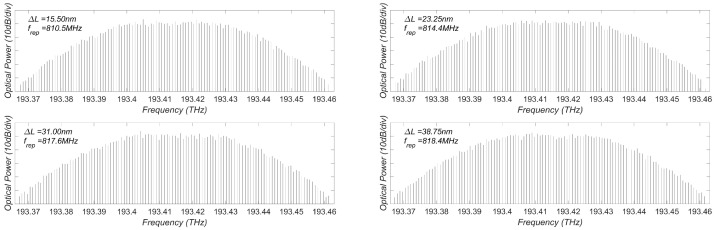
Demonstration of the repetition rate changing with *L*.

**Figure 6 sensors-23-08872-f006:**
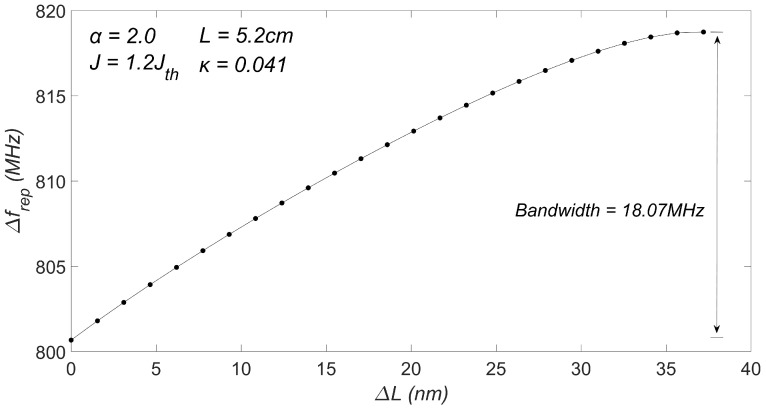
The relationship between ΔL and $\Delta f_{rep}$.

**Figure 7 sensors-23-08872-f007:**
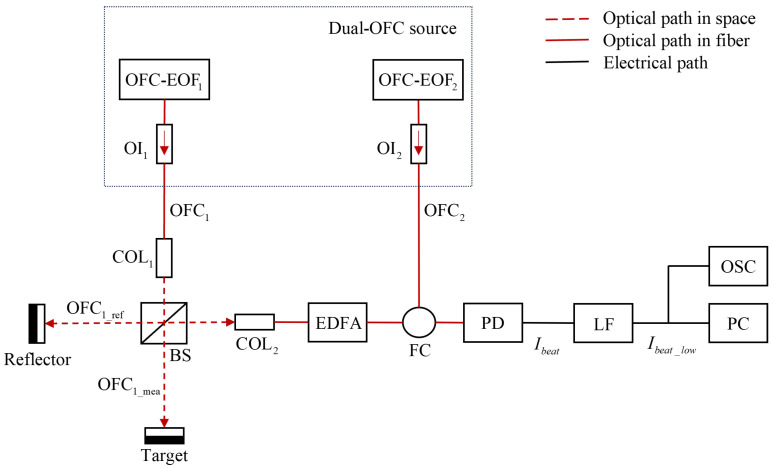
Schematic setup of a dual-OFC system for distance measurement. OFC-EOF, external optical feedback-based OFC sources; OI, optical isolator; COL, collimator; FC, fiber coupler; EDFA, erbium-doped fiber application amplifier; PD, photodiodes; LF, low-pass filter; OSC, digital oscilloscopes; PC, computer for further processing.

**Figure 8 sensors-23-08872-f008:**
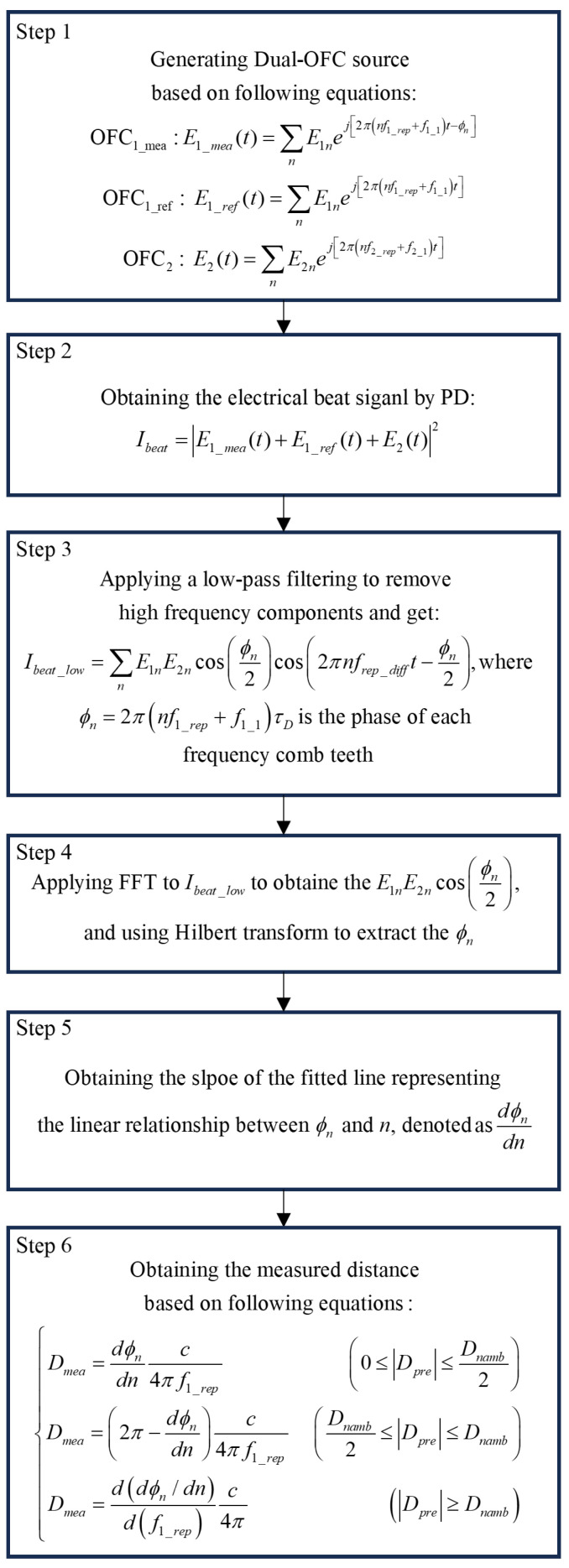
Measurement principle and processing diagram for distance measurement.

**Figure 9 sensors-23-08872-f009:**
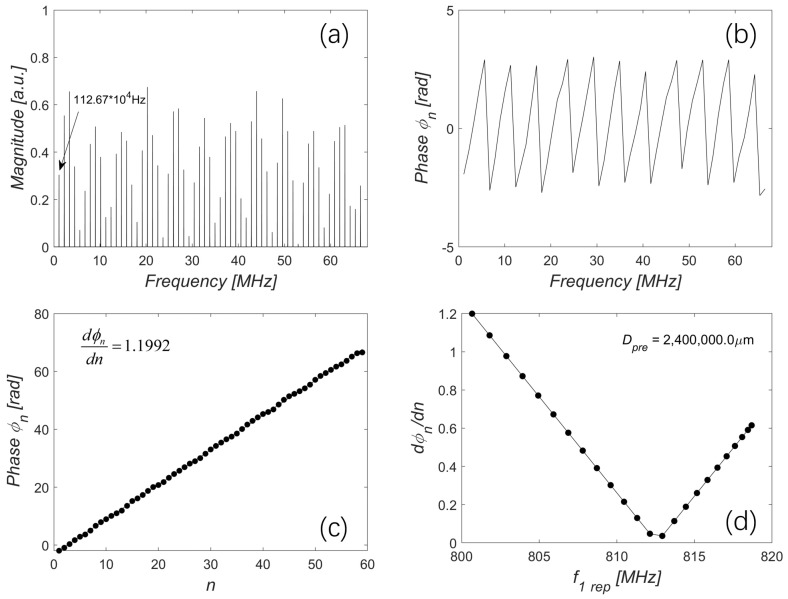
(**a**) Beat spectrum of the $\mathrm{OF}C_{1\;\;\;\_\;\;\;\mathrm{mea}}$ beam, $\mathrm{OF}C_{1\;\;\;\_\;\;\;\mathrm{ref}}$ beam, and $\mathrm{OF}C_{2}$ beam after low-pass filtering. (**b**) Wrapped phase obtained by the Hilbert transform. (**c**) Unwrapped phase and the fit line. (**d**) The relationship between the varied $f_{1\_rep}$ and $d\phi_{n}/dn$ when $D_{pre}$ = 2,400,000.0 μm.

## Data Availability

Data sharing does not apply to this article as no new data were created or analyzed in this study.

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
