# Peer review of "Tunable Optical Frequency Comb Generated Using Periodic Windows in a Laser and Its Application for Distance Measurement"

_sensors, 2023, doi:10.3390/s23218872_

Round 1

Reviewer 1 Report

Comments and Suggestions for Authors

Reviewer’s comments on manuscript Tunable optical frequency comb generated using periodic windows in a laser and its application for distance measurement (Sensors-#2629847)” by Z. Chen et al.

This is a rather well-written theoretical paper – a feasibility study – about distance sensing by a dual frequency comb laser system. Frequency comb lasers represent a fashinable class of new generation lasers promising in metrology, due to their enlarged region of coherence. The treatment is accurate. From my part, I found the error analysis a little short, especially concerning the possible environmental effects (e.g. light scattering) which might affect the applicability – the sensible distance range – very much.  

Subjectal concerns:

1.      Although the high accuracy of the system has been well demonstrated for a given distance, the applicability of the system is not clear, regarding the range of distances which can be measured with tolerable errors. A more detailed error analysis is suggested. Namely, what is the error of distance determination as the function of the value of the actual distance? This function should depend on  enviroment, the applied power, and polarization. The range of applicability can be determined from these curves.         

2.      Also related to the above point. The polarization has not been mentioned at all. It should also be an important factor in determining the range of applicability in different environments. E.g. penetration depth of circularly polarized light is larger than for linearly polarized one, due to the different degrees of light scattering in an environment. Suggested is an extension of the discussion part with a role of polarization, if could be any.     

3.      What is the power requirement for these measurements? It should be discussed.

4.      The definition of Hilbert-transform is missing. Please define it at its first usage.

5.      Can this device – or other one of similar principle – be used in the mm-cm range in medical diagnosis like ultrasound and optical coherence tomography (OCT)? If there could be some medical applications, please discuss them.   

Formal concerns:

1.      Line 84 in Page 2: Please write „form”.

2.      Line 89 in Page 3: Please write „portion”.

3.      Line 188 in Page 6: Please write „for achieving”.

4.      Line 199 in Page 7: Please write „repetition rate”.

In Fig. 9 Please write Panel „c”.

Comments on the Quality of English Language

Spell-checking is recommended!

Author Response

Please see the responses in the attached file.

Reviewer 2 Report

Comments and Suggestions for Authors

Acting like a ruler for light, the optical frequency combs (OFC) are specialized lasers that provide a wide range of stabilized optical wavelengths simultaneously. They measure exact frequencies of light — from the invisible infrared and ultraviolet to visible red, yellow, green and blue light — quickly and accurately. Capitalizing on these inherent advantages, OFCs have found applications in various fields, such as atomic clock comparisons, calibration of astronomical spectrographs, ultra-low-noise microwave generation, distance measurement and laser ranging.

An essential approach to involve leveraging laser dynamics triggered by external perturbations can be utilized to generate OFC. This technique, sometimes referred to as "dynamical OFC", relies on subjecting a laser to an external perturbation, such as a modulation in the driving current or a change in the cavity length. These perturbations cause nonlinear interactions within the laser cavity, leading to the emission of a spectrum of optical frequencies that are evenly spaced, resulting in the OFC.

Compared to other methods of generating OFCs, such as via mode-locked lasers or microresonators, dynamical OFC offers several benefits, including small size, low cost, easy integration and flexibility.

However, it is worth noting that the dynamical OFC approach may have some limitations. For instance, due to the stochastic nature of the perturbations in the laser cavity, the comb accuracy and stability may not be as high as other methods. Furthermore, the implementation of dense OFCs may require more complex and tailored external perturbations.

Nonetheless, the dynamical OFC approach offers a promising solution for generating OFCs, providing a simple, low-cost, and flexible alternative to traditional methods.

In this study, the authors explore the utilization of external optical feedback (EOF) with periodic window to induce laser dynamics for the generation of OFCs with a tunable repetition rate. Additionally, using the tunable repetition rate, a dual-OFC system is proposed for distance measurement, which is commendable. However, before publication, the authors should address the following concerns.

1. In the section of “Introduction”, the authors introduced several approaches to implement OFC. However they did not make a comparison, including the advantages and disadvantages and the main application areas. Therefore, they should make a brief comparison of each OFC technique, especially the advantages and disadvantages of "dynamical OFC".

2. The description of the improvement of traditional "dynamical OFC" technique is not clear and specific, please the authors supplement them.

3. There are no clear experimental parameters, such as the type, model, light intensity and other important parameters of the laser, the length of the cavity parameters, etc., please the authors supplement them.

4. What are the specific meanings of parameters κ, α, J and L? Please add the descriptions.

5. In the ranging experiment, how much is the ranging accuracy? is 0.1um? And what is the corresponding time under the accuracy?

6. As discussed in the introduction of the manuscript, OFCs have found applications in various fields, such as atomic clock  comparisons, calibration of astronomical spectrographs, ultra-low-noise microwave generation, distance measurement and laser ranging. For the authors’ information, I believe it is for sure potentially useful in long distance quantum key distribution, Phys. Rev. A 98, 062323 (2018) ,  Nature volume 557pages400–403 (2018) (very recent QKD experiment exceeding 1000 km, where precise control and monitoring the phase and frequency is needed.

Author Response

(The authors gave the same response as above.)

Round 2

Reviewer 2 Report

Comments and Suggestions for Authors

It is publishable now.